# Recent Progress in Morphology-Tuned Nanomaterials for the Electrochemical Detection of Heavy Metals

**DOI:** 10.3390/nano12223930

**Published:** 2022-11-08

**Authors:** Chinchu Gibi, Cheng-Hua Liu, Scott C. Barton, Jerry J. Wu

**Affiliations:** 1Department of Environmental Engineering and Science, Feng Chia University, Taichung 407, Taiwan; 2Department of Chemical Engineering and Materials Science, Michigan State University, East Lansing, MI 48824, USA

**Keywords:** heavy metals, morphology, dimensions, electrocatalysts, electrochemical sensors, nanomaterials

## Abstract

Heavy metals are one of the most important classes of environmental pollutants which are toxic to living beings. Many efforts are made by scientists to fabricate better sensors for the identification and quantification of heavy metal ions (HMI) in water and food samples to ensure good health. Electrocatalysts have been demonstrated to play an important role in enhancing the sensitivity and selectivity of HMI detection in electrochemical sensors. In this review, we presented morphologically well-tuned nanomaterials used as efficient sensor materials. Based on the molecular dimensions, shapes, and orientation, nanomaterials can be classified into 0-D, 1-D, 2-D, and 3-D nanomaterials. Active surface areas with significant exposure of active sites and adsorption–desorption abilities are extensively varied with dimensionality, which in turn ultimately influence the sensing performance for HMI.

## 1. Introduction

Heavy metals are naturally occurring elements having an atomic number greater than 20 and an elemental density greater than 5 g/cm^3^ [1]. Some of these heavy metals, such as Mn, Fe, Co, Ni, Cu, Zn, and Mo, are essential micronutrients for living organisms. An optimum amount of these essential heavy metal ions (HMI) is necessary for the proper functioning of the human body. Their deficiency as well as abundance can cause diseases. On the other hand, Cd, Pb, and Hg do not have any role in the physiological functions of living organisms and are highly toxic beyond a threshold limit [2]. The permissible limit in the aqueous media of these toxic HMIs as prescribed by the WHO is given in Table 1 [3]. Heavy metals are released into the environment through natural processes, like volcanic eruptions, withering of minerals, forest fires, etc. However, anthropogenic activities such as the wide use of pesticides, unscientific disposal of industrial wastes, combustion of fuels, and mining significantly contribute to HMI pollution [4].

The most dangerous aspect of HMI pollution is due to their nature of bioaccumulation and persistence. The HMIs released into the environment enter into the food chain and continue to be accumulated in the human body at an alarming rate as they are persistent in nature and are not excreted out, resulting in fatal diseases. The Environmental Protection Agency (EPA) and International Agency for Research on Cancer (IARC) in the USA have classified As, Cd, Cr, Pb, and Hg as known or probable human carcinogens [5]. In addition to mutagenic effects, heavy metals are also associated with reproductive toxicity (Ni, Hg, Pb), kidney (Ni, Cd, Hg) and lung (Ni, Cd) disorders, liver damage (Cu, Cd), neurotoxicity (Hg, Tl, Pb, Hg), and Alzheimer’s disease, etc. [5,6] Therefore, there is necessitous demand for efficient sensors to detect heavy metals in real environmental samples.

Conventional methods for the detection of heavy metal include liquid chromatography [7], inductively coupled plasma mass spectroscopy (ICP-MS) [8], atomic absorption spectroscopy (AAS), atomic emission spectroscopy (AES) [9], atomic fluorescence spectroscopy [10], and inductively coupled plasma-optical emission spectrometry (ICP-OES) [11]. Some of these methods are sensitive and selective, but they are expensive, laborious, and time-consuming, which may often not be beneficial for real sample analysis and require highly trained personnel for analysis Electrochemical methods are preferred over conventional methods due to their high sensitivity, selectivity, cost-effectiveness, promptness, portability, simultaneous detection of multiple analytes, and environmental friendliness while sensing HMIs. Several electrochemical methods, such as conductometry, voltammetry, amperometry, and potentiometry, are available for the detection of various analytes. Among these, voltammetry is the most commonly used technique. Voltammetric techniques involve the measurement of current with variation in potential. With the mode of variation in voltage, voltammetric techniques are classified as linear sweep voltammetry (LSV), cyclic voltammetry (CV), differential pulse voltammetry (DPV), and anodic stripping voltammetry (ASV) [12,13,14]. Sometimes a combination of these techniques, such as differential pulse anodic stripping voltammetry (DPASV), square wave anodic stripping voltammetry (SWASV), and linear sweep anodic stripping voltammetry (LSASV), are used for sensing various environmental contaminants [12,14].

Electrochemical sensors are classified into (a) DNA-based, (b) enzyme-based, and (c) electrocatalyst-based based on their recognition element [15]. DNA-based sensors use DNA as the biological recognition element. The interaction between the analyte and DNA can be electrostatic, groove binding, or intercalating. Enzyme molecules are used as the recognition element in enzymatic biosensors. The highly selective nature of enzymes toward particular analytes is used for the fabrication of enzyme-based sensors. In electrocatalyst-based sensors, the electrocatalyst on the electrode surface undergoes a redox reaction with the analyte. This selective interaction is used for sensitive analysis of various environmental contaminants. As compared with the other sensors, electrocatalyst-based sensors are an appropriate tool for the simultaneous detection of multiple analytes. Portability is another advantage of electrocatalyst-based sensors as it is very helpful in field sample analysis.

Electrochemical sensors utilize electrocatalysts as modifications on the working electrodes for catalyzing redox reactions to improve the selective detection of an analyte [16,17]. There has been a booming growth in the use of nanomaterials as electrocatalysts for various applications due to their higher surface area, which enhances the electrocatalytic activity. The advancement of nanotechnology has made it possible to engineer materials with strikingly different shapes and sizes, offering selectivity toward specific analytes. Selectivity of the nanomaterial for the detection of a particular HMI can be improved by incorporating an appropriate functional group with high binding affinity with the analyte [18]. The modifications of electrocatalysts can affect chemical or physical interaction with the HMIs. The characteristics of an electrocatalyst can not only be tuned by the nature of the materials used, but also by their morphology.

The morphology of an electrocatalyst refers to its shape, size, volume, texture, and distribution. Change in morphology can bring out new physical and chemical properties to the same material [19]. The shape of the nanomaterial can influence the electroanalytical features due to their different atom distribution [20,21]. An electrocatalyst with good morphology gives a better electrochemical response by decreasing the adsorption–desorption energy barrier, increasing the surface area, and exposing more adsorption sites on the electrocatalyst for better signal response [22,23,24]. Changes in experimental parameters, such as pH, temperature, precursor concentration, stabilizers, stirring rate, reaction time, etc., can influence the morphology of the nanoparticles used for the modification of the electrode. Different shapes, such as spherical, trigonal, tetrahedral, tube, wire, star, flower, etc. have been reported for the detection of HMIs.

Classification of nanomaterials was primarily performed by Herbert Gleiter based on crystalline forms and chemical composition [25]. He classified nanostructured materials into layer-shaped, rod-shaped, and equiaxed crystallites. However, his scheme of classification was devoid of fullerenes and nanotubes. Later, Pokropivny and Skorokhod classified nanomaterials into zero-dimensional (0-D), one-dimensional (1-D), two-dimensional (2-D), and three-dimensional (3-D) [26]. They also proposed that nanoparticle shapes and dimensionalities greatly affect their characteristics. Quantum dots, fullerenes, noble metal nanoparticles, magnetic nanoparticles, and polymer dots constitute 0-D nanomaterials. Nanowires, nanotubes, and nanoribbons belong to the class of 1-D nanomaterials. Nanoplates, nanoflakes, nanodisks, and nanosheets are some of the 2-D nanomaterials reported for the detection of heavy metals. Nanoflowers, dendrites, nanocubes, nanocages, and nanotetrapods belong to the class of nanomaterials in 3-D regimes. Besides these, nanoparticles with hetero dimensions, i.e., 0D-1D, 1D-2D, and 2D-3D, have also been reported as electrocatalysts for environmental sensing applications. Thus, this review aims to discuss the influence of the morphology of nanomaterials for the detection of HMIs as below.

## 2. Classification of Nanomaterial Morphology Based on Dimensions

### 2.1. Zero-Dimensional (0-D) Nanomaterials

Zero-dimensional (0-D) nanomaterials have been claimed as the antecedent for developments in nanotechnology. It has all three dimensions in nanoscale, i.e., below 100 nm. Electrons in 0-D materials are confined within the nanoscale in all dimensions and are not delocalized [27]. 0-D nanomaterials combine the properties of extremely small size, quantum confinement, edge effects, and good chemical stability [28].

Bhanjana et al. first reported SnO_2_ quantum dots for the detection of Cd ions [29]. Morphological studies revealed that the spherical-shaped quantum dots (Figure 1a) with an average diameter of 3 ± 1 nm were homogeneously distributed, and they acted as electron mediators for the detection of Cd ions. The sensor displayed a detection limit of 0.5 ppm and a sensitivity of 77.5 × 10^2^ nA ppm^−1^ cm^−2^. The corresponding CV for the analysis is given in Figure 1b. The selectivity of the sensor toward Cd ions was clearly visible from the amperometric response as shown in Figure 1c. Sometimes, noble metals were added to improve the conductivity of the 0-D materials. Conjugates of graphene quantum dots and gold nanoparticles (Figure 1d) were synthesized for the simultaneous detection of Hg(II) and Cu(II) ions [30]. The mechanism for the detection of Hg ion using anodic stripping analysis is given in Figure 1e. The synergistic interaction between gold nanoparticles and graphene quantum dots resulted in a low detection limit (0.02 and 0.05 nM) and high sensitivities (2.47 and 3.69 μA/nM) for Hg(II) and Cu(II), respectively. A comparison of the anodic stripping response of Hg(II) and Cu(II) ions, and a mixture of them on GQD-AuNPs /GCE, is shown in Figure 1f, where well separated peaks were observed for Hg(II) and Cu(II) ions and the intensity of the Hg(II) ion peak was reduced in the stripping response of mixture of Hg(II) and Cu(II) ions. However, difficulty in regulating the size of 0-D nanomaterials during synthesis and toxicity associated with some 0-D nanomaterials, such as quantum dots, may limit their use as sensors for the detection of HMIs, especially in real samples.

### 2.2. One-Dimensional (1-D) Nanomaterials

*One*-dimensional (1-D) materials have two dimensions in nanoscale, yielding needle-like shapes. Electrons in 1-D materials are confined within two dimensions, i.e., electrons cannot freely move [27]. Properties of being mechanically flexible and robust at the same time pave the way for continuous and long networks of 1-D nanomaterials with a lower number of grain-based boundaries and much-exposed atoms on the surface for highly sensitive detection of HMIs [31].

Noble metals, such as Au, Ag, Pt, and Pd, are often used as modifications to improve the electron transfer ability and thereby enhance the conductivity of the 1-D nanomaterials. Simultaneous detection of Pb(II), Cd(II), and Cu(II) ions was achieved by using carbon nanotubes decorated with Au nanoparticles as shown in Figure 2a [32]. The excellent stability and good transfer ability of carbon nanotubes helped in the electrochemical performance of the sensor. Also, size-controlled Au nanoparticles on the carbon nanotubes increased the electron transfer ability of pristine carbon nanotubes which further improved the sensing ability. The limit of detection reported was 0.1 μmol/L for the detection of Pb, Cd, and Cu ions with a deposition time of 150 s by SWASV technique. The corresponding voltammogram is given in Figure 2b.

Metal oxides tuned into 1-D morphology often act as a good substrate for the detection of HMIs. Zinc oxide nanopillar-modified gold electrodes (Figure 2c) were utilized for the detection of Cd(II) ions using CV and chronoamperometric techniques [33]. The linear detection ranges of CV and chronoamperometric technique were 5–45 ppm (Figure 2d) and 5–50 ppb, respectively. The sensor showed good selectivity toward Cd(II) ions with a detection limit of 4 ppb.

Another approach was the use of conducting polymers, such as polyaniline and polythiophene, etc., in 1-D morphology. A sensor highly selective toward Hg^2+^ was developed by Narouei et al. using a conductive polymer poly(aniline-co-o-aminophenol), PANOA, with Au nanoparticles homogeneously distributed on its surface [34]. The morphology studies revealed that the formation of Au nanoparticles was homogeneously distributed on the PANOA fiber network (Figure 2e). The presence of nitrogen-functional groups on PANOA was demonstrated along with the Au nanoparticles which have good adsorption for Hg(II) ions and improved the detection ability of the sensor. The synergistic effects of PANOA fiber and Au nanoparticles resulted in a good sensor for Hg(II) ion detection. This material yielded a lower detection limit of 0.23 nM for the detection of Hg^2+^ ions with SWASV analysis. The corresponding voltammogram from which the calibration curve was plotted is given in Figure 2f. The sensor was successfully used in the determination of Hg(II) in water and fish samples.

Recently, nanoclay mineral composites with MWCNTs have been prepared for improving the conductivity of 1-D materials. A mixture of Bi nanoparticles, MWCNTs and nano-size sodium montmorillonite (MWCNT-NNaM) have been synthesized for electrochemical detection of HMIs [35]. The combined effect of better adsorption properties offered by the nano-clay structures and higher electrochemical activities of Bi NPs on MWCNTs resulted in lower detection limits of 0.707 μM,0.097 μM, 0.008 μM, and 0.157 μM for the detection of Zn(II), Cd(II), Pb(II), and Cu(II), respectively. A major challenge with sensors made of 1-D nanomaterials is the controlled growth of very thin and long nanomaterials that render better sensitivity toward the detection of HMIs [36].

### 2.3. Two-Dimensional (2-D) Nanomaterials

Two-dimensional (2-D) materials have one dimension in nanoscale, resulting in plate-like shapes. In 2-D nanomaterials, electrons are confined in one dimension [27]. 2-D materials have a large active surface area. Owing to the high surface area, good mechanical stability, and electrical conductivity, 2-D nanomaterials have been proven to be ideal for sensor applications. Graphene (GR) sheets, graphene oxide (GO) sheets, and reduced graphene oxide (r-GO) sheets are the most commonly used 2-D materials, which have also been used as a support for catalysts in the detection of various contaminants. Recently, a metal-free approach using GR-GO composite sheet was attempted to detect Cd(II) ions in water samples [37]. Enhanced hydrophilicity of GO and good conductivity of GR resulted in a detection limit of 0.087 μM and a linear range of 0–10 μM for the detection of Cd(II) ions. Graphene/CeO_2_ sheets were used for simultaneous and selective detection of Cd(II), Pb(II), Cu(II), and Hg(II) [38]. The presence of a large number of oxygen-containing groups on the surface of graphene oxide sheets provided active centers for the deposition of CeO_2_ nanoparticles. The large surface area of graphene sheets acted as a support for the deposition of CeO_2_ on both sides of the sheet [39]. CeO_2_ nanoparticles enhanced the surface area through the porous structure as seen in Figure 3a. The exposed surface area provided more sites for the adsorption of HMIs and thus improved the catalytic efficiency of the material. The real-time application was tested in wastewater samples using the DPASV technique (Figure 3b) and the detection limit was lower than the WHO guidelines. 

In addition to traditional 2-D graphene sheets, metal oxides can also form 2-D nanostructures for HMI sensing. Li et al. have synthesized a stacked nano plate-like structure of Fe_3_O_4_ using the hydrothermal method for concurrent detection of Zn^2+^, Cd^2+^, Pb^2+^, Cu^2+^, and Hg^2+^ ions [22]. The nanoplate stacked slices of Fe_3_O_4_ had a diameter of 500 nm and a thickness of 2.30 nm (Figure 3c). The sensitivity values for the five ions were found to be quite distinguishable. It was highly sensitive toward Pb^2+^ and least toward Zn^2+^, as shown in Figure 3d. The adsorption studies also followed the same trend, which suggests the linear relationship between adsorption and detection sensitivity. The special morphology of the nanoplate structure with exposed rough edges provided the site for better adsorption and hence offered a lower detection limit and better sensitivity. Liao et al. compared the electrochemical activities of NiCo_2_O_4_ nanoparticles and NiCo_2_O_4_ core-ring nanoplatelets and indicated the role of adsorption and desorption in sensing behaviors [23]. The NiCo_2_O_4_ core-ring nanoplatelets-modified GCE showed 1.7 times higher sensitivity and 2.64 times lower detection limit than NiCo_2_O_4_ nanoparticles for the detection of Pb(II), Cd(II), Hg(II), and Cu(II). The core-ring structure of nanoplatelets provided two times larger surface area than that of NiCo_2_O_4_ nanoparticles. The order of sensitivity for the detection of various HMIs was: Pb(II) > Cd(II) > Hg(II) > Cu(II), which was evidenced using the adsorption and desorption values. The adsorption energy of Pb was maximum and hence accounted for the higher sensitivity of Pb(II) ions.

Often, TMO nanosheets are doped to improve their sensitivities. Ni-doped Co_3_O_4_ nanosheets were synthesized for sensing Hg(II) ions [40]. Optimization of the doping level of Ni augmented the redox activity and oxygen vacancies of porous Co_3_O_4_ nanosheets and revamped the electrochemical conductivity of pristine Co_3_O_4_ nanosheets. Ni/Co_3_O_4_ (NC5.0)/GCE granted a sensitivity of 864.93 μA μM^−1^ cm^−2^ and a detection limit of 0.009 μM for the detection of Hg(II) ions in water samples. 

Recently, two-dimensional materials, MXene, have gained much interest as electrocatalysts in batteries, supercapacitors, and sensors due to stability, good conductivity, and possession of large surface area which are hydrophilic [41,42]. For the first time, Bi nanoparticles were attached to Ti_3_C_2_T_x_ MXene nanosheets for the fabrication of electrochemical sensors for HMIs [43]. Figure 3e shows the two-dimensional layered structure of Ti_3_C_2_T_x_ nanosheets. The homogeneous large surface area of these nanosheets provided the sites required for the adsorption of Bi nanoparticles through electrostatic interactions. Bi nanoparticles were distributed in a scattered way on the surface of Ti_3_C_2_T_x_ nanosheets as shown in the SEM image in Figure 3e. The presence of a large surface area and exposed active sites in Bi nanoparticles further improved the electrocatalytic activity of Ti_3_C_2_T_x_ nanosheets for simultaneous detection of Pb(II) and Cd(II) ions using the SWASV technique as seen in Figure 3f. The practical applicability of these sensors was tested in tap and lake water. Agglomeration of 2-D nanosheets often leads to decreased sensitivity and long response time. Even though 2-D materials act as support, they often fail to provide the good conductivity necessary for sensing. Synthesis of large and very thin nanosheets for better sensitivity is yet another challenge for use of 2-D materials for the detection of HMIs.

### 2.4. Three-Dimensional (3-D) Nanomaterials

Three-dimensional (3-D) materials are not in nanoscale in any dimension i.e., all three dimensions are above 100 nm. Electrons in 3-D nanomaterials are fully delocalized, i.e., electrons freely move in all dimensions [27]. 3-D nanomaterials have unique morphology with a large surface area, improved electrochemical performance, favorable structure stability, and more diffusion sites over 0-D, 1-D, and 2-D nano-catalysts for the detection of HMIs. Several interesting morphologies have been developed to be used as 3-D nanocatalysts for HMI sensing. Various morphologies offer many diffusion sites, which in turn increases the sensitivity for HMI sensing. 

Hybrid structures of different dimensions often form three-dimensional materials. A 3-D graphene-carbon nanotube hybrid material was developed for the simultaneous detection of Cd(II) and Pb(II) ions [44]. A combination of the two-dimensional morphology of graphene oxide sheets and one-dimensional multi-walled carbon nanotubes resulted in an interconnected structure, where the carbon nanotube acted as the conducting platform accelerating the electron transfer from one graphene oxide sheet to another. A detection limit of 0.1 and 0.2 μg/L for Cd and Pb ions, respectively, and a linear range of 0.5–30 μg/L were achieved using the modification. Multi-walled carbon nanotube-enhanced metal-organic frameworks have been developed to detect Cd^2+^ ions in meat samples [45]. An Amine-functionalized Zr(IV) metal-organic framework and UiO-66-NH_2_@MWCNTs composites were synthesized by a one-pot hydrothermal method and showed varying tetrahedral and octahedral cage structures. The synergistic effect between the octahedral structure of UiO-66-NH_2_, rendering high surface area, and MWCNTs with high conductivity are responsible for the excellent electrocatalytic performance toward the detection of Cd^2+^ ions. The limit of detection was found to be 0.2 μg/L. Lu et al. fabricated three-dimensional honeycomb-like nitrogen-doped carbon nanosheet frameworks with Bi nanoparticles embedded in them as shown in Figure 4a through a one-step process for the detection of Pb^2+^ and Cd^2+^ ions [46]. The highly porous, stacked, and interconnected 3-D structure of carbon nanosheets increased the surface area, promoting the mass transfer of the sample, and the entrapped Bi nanoparticles provided active sites for HMIs, enhancing the catalytic activity for the detection of HMIs. Cd^2+^ and Pb^2+^ ions were simultaneously detected with detection limits of 0.02 μgL^−1^ and 0.04 μgL^−1^ and sensitivities of 0.207 μAμgL^−1^ and 0.273 μA μgL^−1^, respectively, by making use of the three-dimensional honeycomb-like structure. The corresponding SWASV curve is shown in Figure 4b.

Dandelion-like polyaniline-coated gold nanocomposites were prepared by a green and one-pot method for the simultaneous detection of Cu(II) and Pb(II) [47]. The dandelion structure has a spherical part with an average diameter of 430 nm and a rod portion of 240 nm diameter and 0.5–1μm length. Several thorns with an average length of 30 nm were found on the spherical surface. The excellent electrochemical performance of this electrocatalyst was attributed to the special morphology of the Au@PANI nanocomposite yielding a high surface area, and thereby more adsorption sites. Also, the amine and imine groups presented in the conductive polymer PANI and the small size of Au nanoparticles give a seat for better adsorption. A detection limit of 0.003 μM for Pb(II) and 0.008 μM for Cu(II) was obtained using this electrocatalyst. In addition, the applicability of the sensor was tested in lake water samples. 

Niu et al. have developed hexagonal prisms with hexagonal pyramid tips of metal-organic framework (MOF) modified V_2_O_5_ nanoparticles (Figure 4c) through a one-pot hydrothermal reaction for electrochemical detection of Pb^2+^ ions [48]. The pore size of the material was found to be 6.93 nm. The excellent selectivity of the sensor toward Pb^2+^ ions is assumed to be due to the highly porous nature of the material and better ion transmission through its uniform mesoporosity. Figure 4d shows the effect of an increase in the concentration of Pb ions with the sensor in DPV analysis. A zeolitic imidazolate framework (ZIF), a class of MOF doped with nitrogen was synthesized for tracing Cd(II) in environmental and tobacco samples [49]. The star shape of the nanomaterial prepared exhibited plentiful active sites, a large surface area, and ample conductive channels for Cd(II) ion detection. The Star ZIF-8-Nafion/GCE yielded a lower detection limit of 0.48 μg/L using the SWASV technique in water samples. 

Recently, butterfly-shaped Ag nanomaterials were developed for the detection of Cd(II), Pb(II), Cu(II), and Hg(II) [50]. The special butterfly-like morphology of Ag nanoparticles improved the surface area for the adsorption of HMIs, thus improving its catalytic activity toward HMI detection. The material provided low detection limits of 0.4 ppb, 2.5 ppb, 7.3 ppb, and 0.7 ppb for Cd(II), Pb (II), Cu(II), and Hg(II), respectively. Lei et al. synthesized different morphologies of carbon-supported transition metal spinel oxides and their sensing performances toward synchronous detection of Pb^2+^ and Hg^2+^ [24]. Three different morphologies, namely blooming flower (Figure 4e), carambola (Figure 4g), and three-dimensional sphere, were observed for Ni–manganate-supported graphene (NMO-GR), zinc–manganate-supported graphene (ZMO-GR), and copper–manganate-supported graphene (CMO-GR), respectively. Their electrochemical responses toward the metal ions were quite different. NMO-GR showed the best electrochemical activity among the three and CMO-GR the least. The high specific surface area offered by the blooming flower and carambola structure of NMO-GR and ZMO-GR exhibited maximum active sites for the enhancement of electron transfer of Pb^2+^ and Hg^2+^ than the spherical CMO-GR. The electrochemical sensing ability of NMO-GR (Figure 4f) and ZMO-GR (Figure 4h) for the detection of Pb and Hg ions were determined and compared using the SWASV technique. The special morphology of NMO-GR provided a lower detection limit of 0.050 nM for Pb^2+^ and 0.027 nM for Hg^2+^; where, using ZMO-GR, the limit of detection obtained was 0.080 nM for Pb^2+^ and 0.040 nM for Hg^2+^.

## 3. Comparison of Sensitivities of Nanomaterials with Different Morphologies

Nanomaterials of the same composition and different morphologies have different sensitivities for HMI detection. The major reasons for this can be ascribed to the difference in surface area and exposed active sites. Improved surface area may offer uninterrupted electron transfer and hence improves the sensitivity. Adsorption and desorption processes also play a huge role in electrochemical HMI sensing. The adsorption and desorption sites can be regulated by calibrating the morphologies of nanomaterials used for sensing.

Three different morphologies of NiO—namely NiO rods, NiO flakes, and NiO balls—were prepared by the hydrothermal method through slight variations in precursors and reaction conditions, and then their abilities to detect HMIs, such as Pb and Cd, were compared [51]. The sensitivity offered by these three morphologies was ranked in the order of rods < flakes < balls. The BET surface area of rods, flakes, and balls were 66.0868, 70.256, and 186.4799 m^2^g^−1^, respectively, where the surface area of balls was three times that of rods, which pave the way for optimal electron diffusion capability in NiO balls. The adsorption capacities also followed the order of rods < flakes < balls. Thus, the high surface area and better adsorption capacity of NiO balls as compared with the NiO rods and flakes resulted in their superior sensitivity of 13.46 AM^−1^ for Pb^2+^ and 5.10 AM^−1^ for Cd^2+^, and a lower detection limit of 0.08 μΜ for Pb^2+^ and 0.07 μΜ for Cd^2+^.

Recently, three different morphologies of CeO_2_, such as nanorod (r-CeO_2_), nanocube(c-CeO_2_), and nanopolyhedra (p-CeO_2_), have been prepared for the simultaneous detection of Cd^2+^ and Pb^2+^ ions [52]. Expanded graphite was used as the carbon support to prevent the aggregation of CeO_2_ nanoparticles. The variation in peak current value is of the order c-CeO_2_/EG/GCE < p-CeO2/EG/GCE < r-CeO_2_/EG/GCE. The active surface area of these electrodes obtained from CV using Randles Sevcik equation were 0.043, 0.097, 0.123, 0.132, and 0.140 cm^2^ for GCE, EG/GCE, c-CeO_2_/EG/GCE, p-CeO_2_/EG/GCE, and r-CeO_2_/EG/GCE, respectively, which followed the same order of peak current value. This implies that the high active surface area of r-CeO_2_ increases the rate of electron transfer. This is also clear from the charge transfer resistance values of electrodes which follow the order c-CeO_2_/EG/GCE > p-CeO_2_/EG/GCE > r-CeO_2_/EG/GCE, which means that faster electron transfer occurs in r-CeO_2_/EG/GCE when compared with other electrodes. The rod-like morphology seems to offer a better sensitivity for heavy metal detection than nanocube or polyhedral morphology. Therefore, the influence of morphology in improving the catalytic activity of the modified electrode was declared. A detection limit of 0.39 and 0.21 μgL^−1^ was achieved for the detection of Cd^2+^ and Pb^2+^, respectively, using r-CeO2/EG/GCE.

Li et al. compared the electrochemical performances of Fe_2_O_3_ nanorods and nanocubes toward Pb(II) ion detection [53]. The two different morphologies of Fe_2_O_3_ showed a varied electrochemical response. The detection limit of nanorods (0.0034 μM) was much lower than that of hollow nanocubes (0.083 μM). Nanorods showed six-fold higher sensitivity than hollow nanocubes for the detection of Pb(II) ions. Despite the large surface area inherited by hollow nanocubes, the low sensitivity was due to its less exposed active sites as compared with the nanotubes of Fe_2_O_3_. The presence of a larger number of active sites in nanotubes can help in better adsorption of Pb (II) ions. Adsorption studies also showed that the amount of Pb(II) ions adsorbed on Fe_2_O_3_ nanorods was 1.9 times that of Fe_2_O_3_ hollow nanocubes. This report suggested the significance of exposed active sites over the surface area for the detection of HMIs. All these comparison studies have suggested that the electroactive surface area is important in any morphology. The surface area of the catalysts can be experimentally determined using BET studies. The presence of exposed active sites for adsorption also determines the ease of adsorption, and thereby the electrochemical detection of HMIs by the specific electrocatalyst is associated with the nature of the material used. The morphologies of nanomaterials for the detection of various HMIs are listed in Table 2 along with the corresponding techniques used, detection limit, linear range, and sensitivities.

## 4. Conclusions and Future Perspectives

Electrochemical sensors have gained great attention in recent years to sense one of the most serious pollutants, heavy metal ions (HMI), in aqueous media. The potential toxicity and bio-accumulating nature of HMI demand versatile sensors to alert us. Several techniques are already available for the detection of these toxic HMI in real samples, where the selectivity, sensitivity, and simultaneous sensing ability of electrochemical sensors should be highly concerned. In this review, we have assessed the performance of morphologically tuned electrocatalysts in HMI sensing. It is concluded that a well-tailored catalyst with a large surface area, exposed active sites, and novel composite material with a better electrochemical sensitivity can deliver better sensor performance. Sometimes, composites of more than one dimension are used for sensing applications to make use of the advantages of different shapes and to decrease the shortcomings of a particular morphology. Possession of a large surface area does not always guarantee better sensing ability. Rather than a large surface area, an increase in active sites for better adsorption can also yield better sensitivity to detect HMIs in real samples. Two main concerns should be further addressed in the future, including the lack of low-cost alternatives to noble metal-modified sensors and unexplored mechanistic aspects of sensing action. The invention and introduction of HMI selective materials with low cost will be feasible once we understand the ambiguity in the sensing mechanism. In addition, the role of parameters, such as shape, directionality, and conducting nature should be explored. Such insight can be a forerunner for the development of better-engineered materials that are inexpensive and resource-efficient for environmental sensing applications. An electrocatalyst with advanced morphology should be explored by avoiding the use of noble metals in most electrocatalytic materials for better conductivity and to be more environmentally benign.

## Figures and Tables

**Figure 1 nanomaterials-12-03930-f001:**
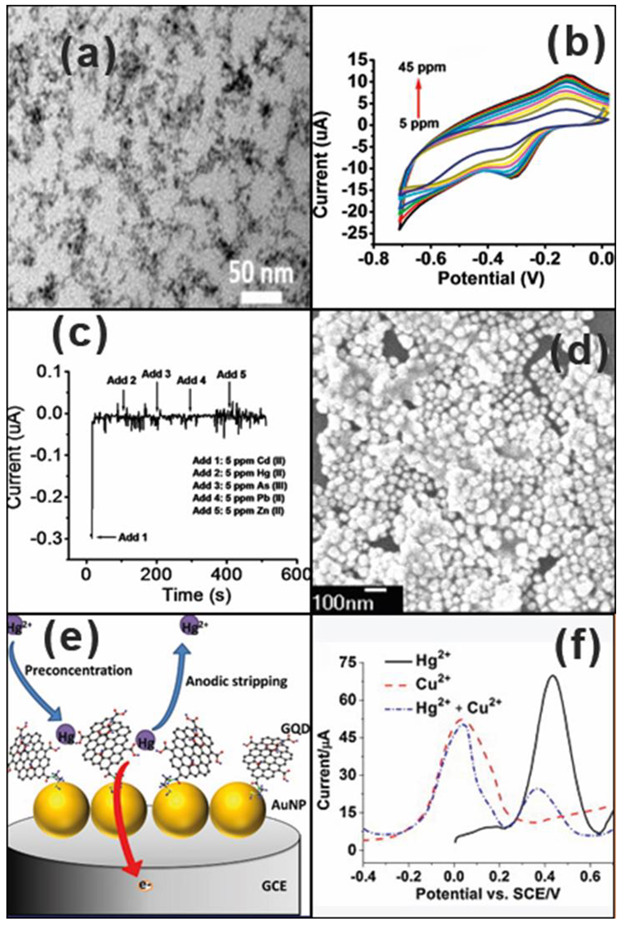
(**a**) HR-TEM image of SnO_2_ QD; (**b**,**c**) CV and selective Amperometric response of SnO_2_/Nafion/Au electrode for detection of Cd ions [29]; (**d**) SEM image of GQD-AuNPs on GCE; (**e**) scheme of Hg^2+^ detection; and (**f**) anodic stripping response of 0.5 μM Hg, 0.5 μM Cu ions, and a mixture of both [30].

**Figure 2 nanomaterials-12-03930-f002:**
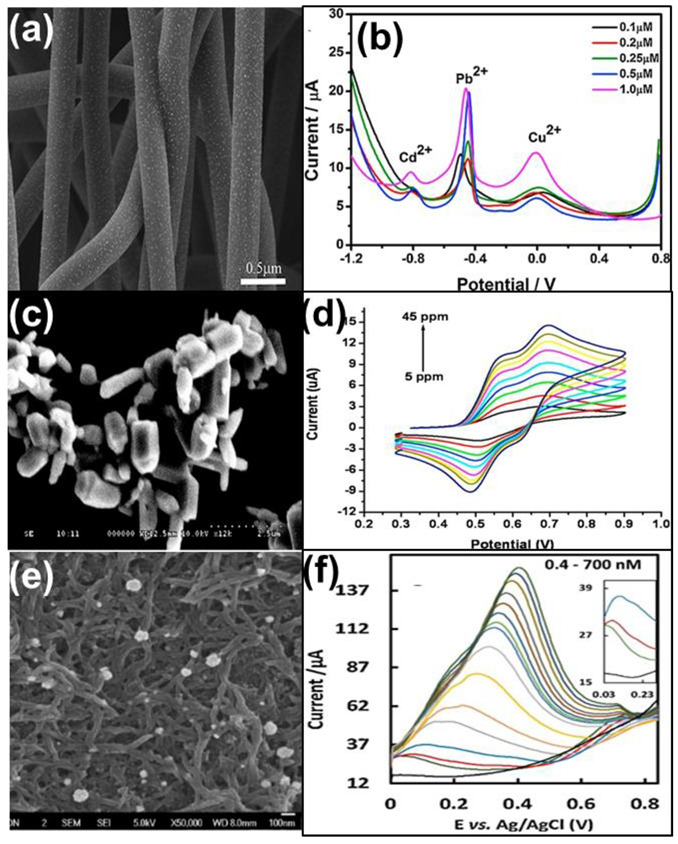
(**a**) FESEM image and (**b**) SWASV of AuNPs/CNFs for simultaneous detection of Cd, Pb, and Cu ions [32]; (**c**) FESEM image of ZnO nanopillar; (**d**) CV of ZnO/Nafion/Au sensor for detection of Cd ions with Ag/AgCl as the reference electrode [32]; (**e**) FESEM image of Au/PANOA; and (**f**) SWASV of Au/PANOA/SPCE for the detection of Hg ions [34].

**Figure 3 nanomaterials-12-03930-f003:**
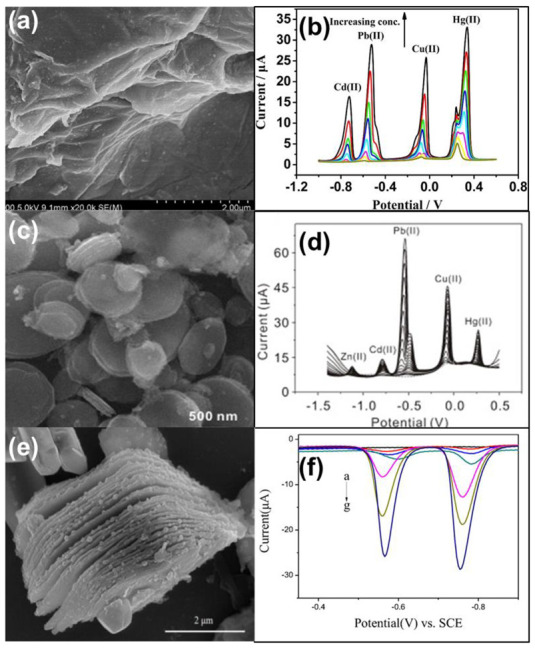
(**a**) FESEM image; (**b**) DPASV of graphene/CeO_2_ hybrid nanocomposite for detection of Cd, Pb, Cu, and Hg ions with saturated Hg/Hg_2_Cl_2_ as the reference electrode [38]; (**c**) SEM image of nanoplate stacked Fe_3_O_4_; (**d**) SWASV of Fe_3_O_4_/GCE for detection of Zn, Cd, Pb, Cu, and Hg ions with Ag/AgCl/KCl (3 M KCl saturated with AgCl) as the reference electrode [22]; (**e**) SEM image; and (**f**) SWASV response of BiNPs@Ti_3_C_2_T_X_ for detection of Pb and Cd ions [43].

**Figure 4 nanomaterials-12-03930-f004:**
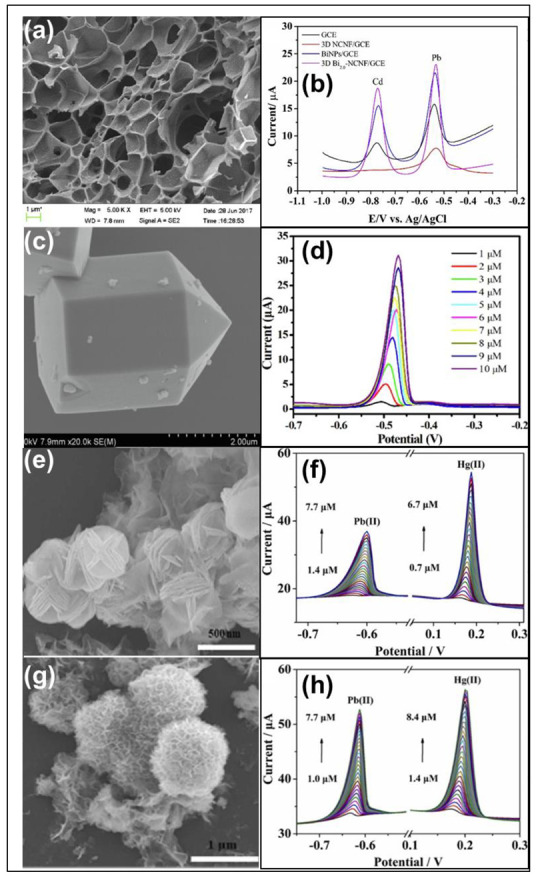
(**a**) SEM image of 3D honeycomb-like structure and (**b**) SWASV curve of Bi_2.0_-NCNF for simultaneous detection of Cd and Pb ions [46], (**c**) SEM image of hexagonal prism with hexagonal pyramid tip, (**d**) DPV curve of V_2_O_5_ @MOF for detection of Pb ions with Ag/AgCl (3 M KCl) as the reference electrode [48], (**e**) SEM image of blooming flower-like structure, (**f**) SWASV of NMO-GR for detection of Pb and Hg ions with Ag/AgCl as the reference electrode, (**g**) SEM image of carambola like structure, and (**h**) SWASV of ZMO-GR for detection of Pb and Hg ions with Ag/AgCl as the reference electrode [24].

**Table 1 nanomaterials-12-03930-t001:** Guidelines for heavy metal ions (HMIs) in drinking water prescribed by the WHO in 2021 [3].

Heavy Metal Ions (HMIs)	Guideline Value (μg/L)
As	10
Cd	3
Cr	50
Cu	2000
Pb	10
Hg	6
Ni	70
Zn	3000

**Table 2 nanomaterials-12-03930-t002:** Improvement of analytical figures of merit with morphology of electrocatalyst based sensors.

Morphology	Electrode	Analyte	Electrochemical Technique	LOD	Linear Range	Sensitivity	Ref.
Quantum dots	SnO_2_ QD/Nafion/Au	Cd(II)	Amperometry	0.5 ppm		77.5 nA/ppm^2^	[29]
Quantum dots	GQD-Au/GCE	Hg(II) Cu(II)	ASV	0.02 nM 0.05 nM		2.47 μA/nM 3.69 μA/nM	[30]
Nanopillars	ZnO/Nafion/Au	Cd(II)	CV	4 ppb	5–45 ppm	10 μA/cm^2^/ppb	[33]
Nanofiber	AuNps/CNF/GCE	Cd(II) Pb(II) Cu(II)	SWASV	0.1 μM 0.1 μM 0.1 μM			[32]
Nanotube Hollow nanocube	Fe_2_O_3_/GCE	Pb(II)	DPV	0.0034 μM 0.083 μM		109.67 μAμM^−1^ 17.68 μAμM^−1^	[53]
Nanofiber	Au/PANOA/SPCE	Hg(II)	ASV	0.23 nM	0.8–12 nM		[34]
Nanotube	BiNP/MWCNT-NNaM/PGE	Zn(II) Cd(II) Pb(II) Cu(II)	SWASV	0.707 μM 0.097 μM, 0.008 μM, 0.157 μM	2.36–40.0; 40.0–180.0 μM 0.32–2.0; 2.0–240.0 μM 0.03–5.0; 5.0–80.0 μM 0.52–10.0; 10.0–40.0 μM		[35]
Nanosheet	GR/GO/GCE	Cd(II)	DPV	0.087 μM	0–10 μM		[37]
Nanosheet	Graphene/CeO_2_/GCE	Cd(II) Pb(II) Cu(II) Hg(II)	DPASV	0.1944 nM 0.1057 nM 0.1636 nM 0.2771 nM	0.02–2.5 μM 0.01–2.5 μM 0.04–1 μM 0.002–0.12 μM		[38]
Nanoplate	Fe_3_O_4_/GCE	Zn(II) Cd(II) Pb(II) Cu(II) Hg(II)	SWASV	0.100 μM 0.213 μM 0.0595μM 0.221 μM 0.0587 μM	0.4–1.8 μM 0.1–2 μM 0.04–20 μM 0.1–1.5 μM 1–8 μM	3.38 μA μM^−1^ 3.85 μA μM^−1^ 33.3 μA μM^−1^ 20.3 μA μM^−1^ 11.4 μA μM^−1^	[22]
Nanoplatelets	Core-ring NiCo_2_O_4_ nanoplatelets/GCE	Pb(II) Cd(II) Cu(II) Hg(II)	SWASV	25.1 nM 104.1 nM 22.7 nM 10.8 nM	0.1–1 μM	45.97 μA/μM 6.41 μA/μM 16.18 μA/μM 58.5 μA/μM	[23]
Nanosheets	Ni/Co_3_O_4_ (NC5.0)/ GCE	Hg(II)	SWASV	0.009 μM	0–1.6 μM	864.93 μA μM^−1^ cm^−2^	[40]
Nanosheets	BiNPs@Ti_3_C_2_T_x_/GCE	Pb(II) Cd(II)	ASV	10.8 nM 12.4 nM	0.06–0.6 μM 0.08–0.6 μM		[43]
Three-dimensional hybrid structure	GR/MWCNTs/Bi	Cd(II) Pb(II)	DPASV	0.1 μg/L 0.2 μg/L	0.5–30 μg/L 0.5–30 μg/L		[44]
Three-dimensional honeycomb	Bi-NCNF/GCE	Cd(II) Pb(II)	SWASV	0.02 μg/L 0.04 μg/L			[46]
Octahedron	UiO66-NH_2_@MWCNTs/GCE	Cd(II)	DPSV	0.2 μg/L	0.5–170 μg/L		[45]
Dandelion	Au@PANI/GCE	Pb(II) Cu(II)	SWASV	0.003 μM 0.008 μM	0.02–0.72 μM 0.08–2.4 μM		[47]
Hexagonal prisms	V_2_O_5_@MOF	Pb(II)	DPASV	28.9 nM	1–10 μM		[48]
Star shape	Star ZIF-8-Nafion/ GCE	Cd(II)	SWASV		0.5–30 μg/L 30–230 μg/L	0.48 μg/L	[49]
Butterfly shape	AgNS/SPCE	Cd(II) Pb(II) Cu(II) Hg(II)	DPSV	0.4 ppb 2.5 ppb 7.3 ppb 0.7 ppb	5–300 ppb 5–300 ppb 5–500 ppb 5–100 ppb		[50]
Blooming flower carambola	ZMO-GR/GCE NMO-GR/GCE	Pb(II) Hg(II)	SWASV	0.080 nM 0.040 nM 0.050 nM 0.027 nM	1.0–7.7 μM 1.4–8.4 μM 1.4–7.7 μM 0.70–6.7 μM	2.49 μA/μM 4.80 μA/μM 3 μA/μM 5.56 μA/μM	[24]
Nanorods Nanoflakes Nanoballs	NiO/GCE	Pb(II) Cd(II)	SWASV	0.2 μM 0.084 μM 0.07 μM	0.2–1.2 μM 0.1–1 μM 0.2–1 μM	2.87 A/M 4.67 A/M 5.10 A/M	[51]
Nanorods	r-CeO_2_/EG/GCE	Cd(II) Pb(II)	DPV	0.39 μg/L 0.21 μg/L	5–100 μg/L 100–600 μg/L		[52]

## Data Availability

Data can be available upon request from the authors.

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
