# Peer review of "Recent Progress in Morphology-Tuned Nanomaterials for the Electrochemical Detection of Heavy Metals"

_nanomaterials, 2022, doi:10.3390/nano12223930_

Round 1

Reviewer 1 Report

This paper reports a specific aspect of nanomaterials in electrochemical applications, that is the detection of heavy metals, a “classic” class of analytes from the electroanalytical point of view. The paper focuses on recent papers, giving a useful view on this topic. I have just a few remarks and suggestions for the final version of the paper.

1.  Title: “Recent progress” should be converted in the plural form, “recent progresses”.

2. To be rigorous, As (table 1, line 45) is not a metal, it is a metalloid. Thus, if the authors (reasonably) include it in the heavy metals, the first sentence (line 25-26) should be modified: “Heavy metals are naturally occurring metals AND METALLOIDS having atomic number greater than 20 and an elemental density greater than 5 g/cm3“, or “Heavy metals are naturally occurring ELEMENTS having atomic number…”.

3.  Line 64-65 (voltammetric techniques, refs 12,13): in the context of references for electrochemical techniques a book should be cited as well. For instance, “Electroanalytical Methods, F. Scholz Ed., Springer, 2010”, or “D Pletcher, R Greff, R Peat, L M Peter, J Robinson, Instrumental Methods in Electrochemistry, Woodhead Publishing 2001” are adequate choices, or something like these, according to Authors’ preferences.

4.  Line 88-89: the sentence is not completely clear, it should be modified (“The electrocatalysts can have chemical or physical interaction with the HMIs”, or “The modification of electrocatalysts can affect chemical or physical interaction with the HMIs”).

5.  Line 122: actually, fig 1b and 1c do not display detection limit and sensitivity, in this context the calibration curve from ref 25 should be reported. Fig 1c displays selectivity, and this is surely a further feature to report.

The same happens for all the figures (e.g. figure 1 f, and so on) in the text: sensitivity and detection limit are connected to the image of the voltammogram, but it is not correct.

6.  Line 136, caption to figure 1e: concentrations of Hg and Cu ions should be here reported from ref 26.

7.  Line 138: actually, two different definitions can be found about 1-D (and 2-D nanomaterials). The first one (followed also by the authors of this paper) states that one-dimension (1D) nanomaterials have one of their dimensions less than 100 nm (e.g. E.I. Akpan, X. Shen, B. Wetzel, K. Friedrich. Design and Synthesis of Polymer Nanocomposites, in “Polymer Composites with Functionalized Nanoparticles Synthesis, Properties, and Applications Micro and Nano Technologies” 2019, Pages 47-83, https://doi.org/10.1016/B978-0-12-814064-2.00002-0). Conversely, according to the second definition, one-dimensional (1D) nanomaterials are in nanoscale in two dimensions, and one dimension is out of the nanoscale (e.g. Sajid Bashir, Jingbo Liu. Nanomaterials and Their Application, in Advanced Nanomaterials and their Applications in Renewable Energy, Chapter 1, 2015, Pages 1-50, https://doi.org/10.1016/B978-0-12-801528-5.00001-4). Further references can be found in both cases. Anyway, in my opinion the second definition is more coherent, and agrees with the 0-D and 3-D definitions reported in the present paper. The authors can decide to follow the first one, of course, but they should add a reference supporting their choice.

8.  Line 329, “HMI ion sensing”: the term “ion” should be delected, it is included in HMI.

9.  Line 345, “variation in peak current value and is of the order”: maybe something was missed, or “and” has to be deleted.

10.   Line 387: “composites of more than one dimension ARE used” (not “is”). 

11.  Line 444, ref 18: the reference is incomplete, the title of the book including this chapter should be added.

Author Response

Comment 1: Title: “Recent progress” should be converted in the plural form, “recent progresses”.

Response: Title: “Recent progress” is converted in the plural form, “recent progresses”.

Comment 2: To be rigorous, As (table 1, line 45) is not a metal, it is a metalloid. Thus, if the authors (reasonably) include it in the heavy metals, the first sentence (line 25-26) should be modified: “Heavy metals are naturally occurring metals AND METALLOIDS having atomic number greater than 20 and an elemental density greater than 5 g/cm3“, or “Heavy metals are naturally occurring ELEMENTS having atomic number…”.

Response: The first sentence (line 30-31 in the revised manuscript) is modified as “Heavy metals are naturally occurring elements having atomic number greater than 20 and an elemental density greater than 5 g/cm3” and reference [1] is changed as Tchounwou, P; Yedjou, C; Patlolla, A; Sutton D; Heavy Metal Toxicity and the Environment. In Molecular, Clinical and Environmental Toxicology. Experientia Supplementum;Luch, A; Eds.; Springer, Basel, Berlin, Germany, 2012; Vol. 101, 133-164.

Comment 3: Line 64-65 (voltammetric techniques, refs 12,13): in the context of references for electrochemical techniques a book should be cited as well. For instance, “Electroanalytical Methods, F. Scholz Ed., Springer, 2010”, or “D Pletcher, R Greff, R Peat, L M Peter, J Robinson, Instrumental Methods in Electrochemistry, Woodhead Publishing 2001” are adequate choices, or something like these, according to Authors’ preferences.

Response: Lines 73,76 (in the revised manuscript) in the context of references for electrochemical techniques the book, D Pletcher, R Greff, R Peat, L M Peter, J Robinson, Instrumental Methods in Electrochemistry, Woodhead Publishing 2001, as per the reviewer’s suggestion is included as [14].

Comment 4: Line 88-89: the sentence is not completely clear, it should be modified (“The electrocatalysts can have chemical or physical interaction with the HMIs”, or “The modification of electrocatalysts can affect chemical or physical interaction with the HMIs”).

Response: Line 97-98 (in the revised manuscript): The sentence is modified as” The modifications of electrocatalysts can affect chemical or physical interaction with the HMIs.”

Comment 5:  Line 122: actually, fig 1b and 1c do not display detection limit and sensitivity, in this context the calibration curve from ref 25 should be reported. Fig 1c displays selectivity, and this is surely a further feature to report.

The same happens for all the figures (e.g. figure 1 f, and so on) in the text: sensitivity and detection limit are connected to the image of the voltammogram, but it is not correct.

Response: The voltammograms are separately mentioned to avoid the discrepancy between detection limits and the corresponding voltammograms in figures 1(b) (line 138-140), fig 1 (f) (in line 147-148), fig 2(b) (line 177-178), fig 2(f) (line 195-196), fig 4(b) (line 322) and fig 4(d) (line 339-340). The selectivity shown by fig 1(c) is mentioned (in line 138-140).

Comment 6: Line 136, caption to figure 1e: concentrations of Hg and Cu ions should be here reported from ref 26.

Response: Line 159 (in the revised manuscript): caption to figure 1e: concentrations of Hg and Cu ions are reported from the corresponding reference [30] (in the revised manuscript) as 0.5 μM Hg, 0.5 μM Cu ions, and a mixture of both.

Comment 7: Line 138: actually, two different definitions can be found about 1-D (and 2-D nanomaterials). The first one (followed also by the authors of this paper) states that one-dimension (1D) nanomaterials have one of their dimensions less than 100 nm (e.g. E.I. Akpan, X. Shen, B. Wetzel, K. Friedrich. Design and Synthesis of Polymer Nanocomposites, in “Polymer Composites with Functionalized Nanoparticles Synthesis, Properties, and Applications Micro and Nano Technologies” 2019, Pages 47-83, https://doi.org/10.1016/B978-0-12-814064-2.00002-0). Conversely, according to the second definition, one-dimensional (1D) nanomaterials are in nanoscale in two dimensions, and one dimension is out of the nanoscale (e.g. Sajid Bashir, Jingbo Liu. Nanomaterials and Their Application, in Advanced Nanomaterials and their Applications in Renewable Energy, Chapter 1, 2015, Pages 1-50, https://doi.org/10.1016/B978-0-12-801528-5.00001-4). Further references can be found in both cases. Anyway, in my opinion the second definition is more coherent, and agrees with the 0-D and 3-D definitions reported in the present paper. The authors can decide to follow the first one, of course, but they should add a reference supporting their choice.

Response: Lines 163-165,218-219 (in the revised manuscript): Definition of 1-D and 2-D is changed according to Sajid Bashir, Jingbo Liu. Nanomaterials and Their Application, in Advanced Nanomaterials and their Applications in Renewable Energy, Chapter 1, 2015, Pages 1-50 [27] (in the revised manuscript), as: One dimensional (1-D) materials have two dimensions in nanoscale, yielding needle like shapes. Electrons in 1-D materials are confined within two dimensions i.e; electrons cannot move freely (lines 163-165) and Two dimensional (2-D) materials have one dimension in nanoscale, resulting in plate like shapes. In 2-D nanomaterials, electrons are confined in one dimension. (lines 218-219). Definition of 0-D materials is also modified according to [27] as. It has all the three dimensions in nanoscale i.e., below 100 nm. Electrons in 0-D materials are confined within the nanoscale in all dimensions and are not delocalized (lines 130-132). Definition of 3-D is also modified as Three-dimensional (3-D) materials are not in nanoscale in any dimension i.e. all three dimensions are above 100 nm. Electrons in 3-D nanomaterials are fully delocalized i.e. electrons move freely in all dimensions (lines 291-293) according to [27].

Comment 8: Line 329, “HMI ion sensing”: the term “ion” should be deleted, it is included in HMI.

Response: Line 383 (in the revised manuscript), “HMI ion sensing”: is changed to HMI sensing.

Comment 9:  Line 345, “variation in peak current value and is of the order”: maybe something was missed, or “and” has to be deleted.

Response: Line 399-400 (in the revised manuscript), “variation in peak current value and is of the order” is changed to variation of peak current value is of the order.

Comment 10: Line 387: “composites of more than one dimension ARE used” (not “is”). 

Response: Line443:  Composites of more than one dimension “is” used is corrected as composites of more than one dimension are used

Comment 11: Line 444, ref 18: the reference is incomplete, the title of the book including this chapter should be added.

Response: The title of the book ‘Comprehensive Materials Processing’ including the chapter ‘Review of recent developments in sensing materials’ is added as reference [19].(in revised manuscript) as Venkatanarayanan, A.; Spain, E..Review of recent developments in sensing materials. In Comprehensive Materials Processing, Hashmi, S, Batalha, G, Tyne. C., Yilbas, B. Eds, Elsevier, Dublin, Ireland, 2014, Vol. 13, 47-101.

Reviewer 2 Report

The article under review provides a brief introduction and overview in the filed heavy metal ion (HMI) detection with the use of nanomaterials (NMs). The key-point of the article is the analysis of the role of NMS morphology in the HMI detection.  The collected in the article material is worth to be published.

Some issues however need to be clarified before making the final decision on the publication.

1. Herbert Gleiter’ name and his works have to be mentioned when describing the classification of NMs.

2.  The text in the article should be subdivided into paragraphs  for better reading.

3. The meaning of the sentence «Two dimensional (2-D) materials have double dimensions at nanoscale…» is misleading.

I recommend publishing this review article, considering the above-mentioned issues.

Author Response

Comment 1: Herbert Gleiter’ name and his works have to be mentioned when describing the classification of NMs.

Response: Lines 111-117 (in the revised manuscript), Herbert Gleiter’ name and his works are mentioned [25] when describing the classification of NMs. Works of Pokropivny and Skorokhod have also been mentioned [26] as a continuation of Geiters work as given here: Classification of nanomaterials was done primarily by Herbert Gleiter on the basis of crystalline forms and chemical composition [25]. He classified nanostructured materials into layer shaped, rod shaped and equiaxed crystallites. But, his scheme of classification was devoid of fullerenes and nanotubes. Later, Pokropivny and Skorokhod classified nanomaterials into zero- dimensional(0-D), one-dimensional(1-D), two-dimensional(2-D) and three-dimensional(3-D) [26]. They also proposed that nanoparticle shapes and dimensionalities greatly affect their characteristics.

Comment 2:  The text in the article should be subdivided into paragraphs for better reading.

Response: The text in the article has been subdivided into paragraphs for better reading in the revised manuscript.

Comment 3: The meaning of the sentence «Two dimensional (2-D) materials have double dimensions at nanoscale…» is misleading.

Response: The definition of 2-D materials is changed to Two dimensional (2-D) materials have one dimension in nanoscale, resulting in plate like shapes. In 2-D nanomaterials, electrons are confined in one dimension. (lines 218-219)  by quoting reference [27] in the revised manuscript.

Reviewer 3 Report

The present manuscript reviews the state-of-art of the electrochemical detection of heavy metal ions by using nanomaterials in sensors.
The topic cannot be considered completely new, since in the past several publications have dealt with the electrochemical determination of heavy metal ions, but the present work is novel in that it intends to report the results published in the literature by grouping the nanoparticles used in sensors according to their morphology. This approach can be useful for researchers working in this field, therefore, I think that the manuscript is worth to be published.

However, there are some points to be addressed before accepting the manuscript:
l. 138-139: According to the authors: "One dimensional (1-D) materials have a single dimension in the nanoscale, yielding needle like shapes."
Unfortunately, this is not true, since according to the usual classification in one-dimensional nanomaterials (1D), one dimension is outside the nanoscale (or two dimensions are at the nanoscale and one dimension at the macroscale).

l. 183-184: The following is stated in the text: "Two dimensional (2-D) materials have double dimensions at nanoscale, resulting in plate like shapes."
Unfortunately this is also wrong since in two-dimensional nanomaterials (2D) two dimensions are outside the nanoscale (or one dimension is at the nanoscale and two dimensions at the macroscale).

Fig. 1. (f): The quality of the image is poor.

Figs. 2. (b),(d),(f): The quality of the images is poor. The size of the letters and numbers is too small. (d) The reference electrode should be given.

Fig. 3. (b),(d),(f): The quality of the images is poor. (b),(d): The reference electrode should be given. (f): The size of the numbers is too small.

Fig. 4. (b),(d),(f),(h): The quality of the images is poor and the size of the letters and numbers is too small. (d),(f),(h):  The reference electrode should be given.

References: Please check refs. 26 and 49.

Author Response

Comment 1: 138-139: According to the authors: "One dimensional (1-D) materials have a single dimension in the nanoscale, yielding needle like shapes." Unfortunately, this is not true, since according to the usual classification in one-dimensional nanomaterials (1D), one dimension is outside the nanoscale (or two dimensions are at the nanoscale and one dimension at the macroscale). 

Response: Lines 163-165 (in the revised manuscript), Definition of 1-D material is altered according to Sajid Bashir, Jingbo Liu. Nanomaterials and Their Application, in Advanced Nanomaterials and their Applications in Renewable Energy, Chapter 1, 2015, Pages 1-50[27]  as: One dimensional (1-D) materials have two dimensions in nanoscale, yielding needle like shapes. Electrons in 1-D materials are confined within two dimensions i.e; electrons cannot move freely.

Comment 2: Lines 183-184 (in the revised manuscript): The following is stated in the text: "Two dimensional (2-D) materials have double dimensions at nanoscale, resulting in plate like shapes."
Unfortunately this is also wrong since in two-dimensional nanomaterials (2D) two dimensions are outside the nanoscale (or one dimension is at the nanoscale and two dimensions at the macroscale).

Response: Lines 218-219, The definition of 2-D materials is changed as “Two dimensional (2-D) materials have one dimension in nanoscale, resulting in plate like shapes. In 2-D nanomaterials, electrons are confined in one dimension by quoting ref [27].”

Comment 3: Fig. 1(f): The quality of the image is poor. 
Response: The quality of the image is improved in the revised manuscript.

Comment 4: Figs. 2.(b),(d),(f): The quality of the images is poor. The size of the letters and numbers is too small. (d) The reference electrode should be given.

Response: Figs. 2.(b),(d),(f): The quality of the images is improved. The size of the letters and numbers is made as normal. (d)The reference electrode is mentioned in caption to the figure as Ag/AgCl as the reference electrode from ref. [32] (line 212-213) in the revised manuscript.

Comment 5: Fig. 3.(b),(d),(f): The quality of the images is poor. (b),(d): The reference electrode should be given. (f): The size of the numbers is too small.

Response: Fig. 3.(b),(d),(f): The quality of the images is improved. (b),(d): The reference electrode are mentioned as saturated Hg/Hg2Cl2 as the reference electrode  from ref. [38] (line 283) and Ag/AgCl/KCl (3 M KCl saturated with AgCl ) as the reference electrode from ref. [22 ](line 285-286). (f): The size of the numbers is increased.

Comment 6: Fig. 4.(b),(d),(f),(h): The quality of the images is poor and the size of the letters and numbers is too small. (d),(f),(h):  The reference electrode should be given. 

Response: Fig. 4.(b),(d),(f),(h): The quality of the images is improved and the size of the letters and numbers is increased. (d),(f),(h):  The reference electrode are mentioned in figure caption as Ag/AgCl (3 M KCl) as the reference electrode (line 372) from ref.[48], Ag/AgCl as the reference electrode (line 374) and Ag/AgCl as the reference electrode (line 375-376) from ref. [24].

References: Please check refs. 26 and 49.

Response: [26] appears as [30] in the revised manuscript, in the reference format of manuscript as Ting, S.; Ee, S.; Ananthanarayanan, A.; Leong, K.; Chen, P. Graphene quantum dots functionalized gold nanoparticles for sensitive electrochemical detection of heavy metal ions. Electrochim. Acta 2015, 172, 7–11.

[49] appears as [53] in revised manuscript, in the reference format of manuscript as: Li, S.; Zhou, W.; Jiang, M.; Li, L.; Sun, Y.; Guo, Z.; Liu, J.; Huang, X. Insights into diverse performance for the electroanalysis of Pb(II) on Fe2O3 nanorods and hollow nanocubes: Toward analysis of adsorption sites.Electrochim. Acta 2018, 288, 42-51.

Round 2

Reviewer 3 Report

The authors improved the paper in the revised version based on the review comments. I suggest to accept the revised manuscript for publication.